# "Trying to remain calm…but I do reach my limit sometimes": An exploration of the meaning of gentle parenting

**Anne E. Pezalla** [1]*, **Alice J. Davidson** [2]

**1** Macalester College, St. Paul, Minnesota, United States of America, **2** Rollins College, Winter Park, Florida, United States of America

* apezall1@macalester.edu

**Data Availability Statement:** All relevant data are within the manuscript and its Supporting information files.

**Funding:** Annie Pezalla received a $1250 faculty fund from Macalester College for this study. She

## Abstract

Raising young children has always been hard, but evidence suggests that it may be getting harder. The isolation of the pandemic, the pressures to fulfill exacting parenting standards, and the explosion of "expert" parenting advice on social media have fueled the rise of "gentle parenting," an approach that pivots away from older, discipline-heavy parenting typologies and which promises the development of happier, healthier children. Despite the popularity of gentle parenting, it has received no empirical scrutiny. The current study represents the first systematic investigation of what gentle parenting entails. Data were gathered from a sample ($N = 100$) of parents of at least one child between the ages of 2 and 7 from the Midwest, Southeast, and Southwest. Approximately half ($n = 49$) of the sample identified as "gentle parents." Inductive analyses identified this approach as one that emphasizes high levels of parental affection and parents' and children's emotion regulation. Gentle parenting appears to be distinct from other established measures of parenting approaches in its emphasis on boundaries, yet the enactment of those boundaries is not uniform. Overall, gentle parents reported high levels of parenting satisfaction and efficacy, but a subset of gentle parents who were highly critical of themselves reported significantly lower levels of efficacy than the rest of the sample. Statements of parenting uncertainty and burnout were present in over one-third of the gentle parent sample. Implications are discussed for future research and increased support for those who identify as gentle parents.

## Introduction

The COVID-19 pandemic left many parents around the world with a rare gift: less rushing around and more intentional time at home with their children [1]. For many others, though, the pandemic was unilaterally bad. Evidence suggests that, for millions of parents, the pandemic increased parental stress [2] to levels that have not yet dropped to pre-pandemic levels [3] and lowered parents' feelings of efficacy in their parenting [4]. Given the nearly impossible task of simultaneously juggling work responsibilities with caring for children at home due to the closure of schools and childcare facilities, it is no wonder that many parents of young

used those funds to compensate participants. Macalester College, which funded $1250 for participant compensation, had no role in the study design, data collection and analysis, decision to publish, or in preparation of the manuscript.

**Competing interests:** The authors have declared that no competing interests exist.

children experienced poor mental health [5] and reported feelings of stress and exhaustion [6] during the pandemic. Those particularly vulnerable to the pandemic have been high-conflict families [1, 7], families who employed harsh or inconsistent parenting [8], and families with children who were prone to stress or anxiety [9].

Evidence suggests that even beyond the pandemic, the work of rearing children is simply seen as more stressful and difficult than it was in the past. A 2023 study within the United States by the Pew Research Center [10] found that 41% of parents reported that being a parent is tiring, and 29% said it is stressful all or most of the time, with the youth mental health crises reported as a common concern among parents. Within the Pew report, 70% of mothers and 60% of fathers acknowledged that being a parent today is more demanding than it was just a few decades ago.

This spotlight on the suffering of parents is relatively new. Developmental Psychology, as a whole, has long focused on children [11], with parents "instrumentalized at the service of their children's development and best interest" (p. 8). Yet a focus on parents in their own right is merited, especially considering that *parental burnout*, which goes beyond the normality of parental stress and which severely and chronically overwhelms parents' resources to cope, is on the rise [12]. Parental burnout appears to be more prevalent among parents who aim to be "perfect" [13], who are neurotic [14], and who have less social support [15]. The question, then, is *what are these parents doing*?

To understand the current zeitgeist around parenting young children, it is helpful to consider the extent to which contemporary parenting approaches relate to arguably the most commonly-cited framework in parenting scholarship: Diana Baumrind's approaches. Baumrind's [16] theory-derived parent classification system, organized around the axes of parenting warmth and parenting discipline, originally generated three distinct approaches to parenting: authoritative parenting was characterized by high parental warmth and discipline; authoritarian parenting was characterized by low parental warmth and high discipline; and permissive parenting was characterized by high parental warmth and low discipline. Maccoby and Martin [17] later expanded on Baumrind's three original parenting styles by adding a fourth type, neglectful parenting, which characterized parents who were low in both warmth and discipline. Broadly, investigations on these parenting styles highlight authoritative parenting as the optimal style for parents for its associations with wellbeing in children [18, 19]. Optimal parenting, according to this framework, is seen in those who employ lots of warmth and sufficient discipline and structure in their interactions with their children.

Baumrind's typology, however, has not been immune from criticism, namely, that it is Eurocentric, endorsing values that are limited to a primarily white, middle-class population [20], and that four relatively distinct parenting styles do not capture the nuance of parents' lives, since most parents inevitably model a mixture of parenting styles [21]. Gottman et al. [22] critiqued the parenting styles, noting that they provide little structure to parents in specific parent-child interactions. For example, what is a parent to do when a child throws a tantrum over a toy they cannot have? In such scenarios, and in many others, Baumrind's authoritative style of high warmth and structure provides no clear guidelines. To address those limitations, Gottman and colleagues [22] proposed an alternative model that focuses on "emotion coaching," which emphasizes parents' explicit labeling of those emotions, awareness of and empathy for their child's emotions, and problem solving with their child to deal with upsetting situations. Guided by the tenets of emotion coaching, a parent whose child is in the throes of a tantrum could address the child's feelings (e.g., "you look very upset right now"), empathize with the child (e.g., "I've felt the same way as you before"), and then work on problem solving (e.g., "you can't have that toy right now but maybe after we eat dinner"). Outcomes relating to parents' use of emotion coaching have been largely positive, with emotion-coached children

experiencing better self-regulation skills and less stress, although again, most of the focus has been on children [e.g., 23], and predominantly White European participants [24].

Still other parenting approaches include "positive parenting," which emphasizes consistent and unconditional empathy and guidance [25], "conscious parenting," [26], which focuses more on parents' need for introspection and mindfulness in viewing their children as teachers of important life lessons, and most recently, "gentle parenting," [27] which appears to be a catchall for variations of both "positive parenting" and "conscious parenting" and which centers on acknowledging a child's feelings and motivations behind challenging behavior, rather than correcting the behavior itself. Gentle parenting also prioritizes the notion of boundaries and giving children choices rather than orders. Gentle parenting appears to have arisen, at least in popular literature, as a rejection of the discipline- and hierarchical-laden nature of Baumrind's authoritative parenting typology. Speculations have also been shared that gentle parenting may be the embodied rejection of how parents were themselves raised when they were children. Indeed, the recent Pew Research Center [10] report found that 44% of respondents said they wanted to raise their children differently from their own upbringing, with many of those respondents reporting that they would be more gentle and less punitive.

Further complicating the array of parenting strategies and philosophies from which contemporary parents may choose, is the fact that, unlike the above-mentioned parenting typologies, the idea of gentle parenting was not born of scholarship. Indeed, this approach seems to be fueled in great part by parenting "experts" and influencers on social media and YouTube. This reliance on social media platforms appears to have increased during the pandemic [28], and although evidence suggests that social media sources can provide helpful information about caregiving and can boost feelings of social connectedness to other parents [28, 29], without discernment about the validity of parenting claims from so-called "experts," the consumption of this online content can be problematic, leading to heightened feelings of isolation and depression [30], especially among parents with higher anxiety [31]. Child development researchers have not investigated the perspectives of gentle parents to determine a common definition of this approach, how this approach is related to other parenting typologies, the ways that gentle parents respond to specific challenges, and the sustainability of this approach from the perspective of the gentle parents themselves. Toward those ends, this exploratory study was guided by the following four research questions:

1. How do those who identify as "gentle parents" define such an approach?
   Based on popular psychology descriptions of this new approach [27], we anticipated that "gentle parents" would have similar priorities as the "positive parenting" [25] and "conscious parenting" approaches [26], namely, emphasizing empathy and mindfulness.

2. How does "gentle parenting" relate to Baumrind's parenting typology?
   Considering that "gentle parenting" seems to prioritize affection and to disavow punishment, we expected this approach to be more closely related to Baumrind's [16] authoritative and permissive parenting styles than to authoritarian or neglectful styles.

3. How do gentle parenting ideologies relate to parenting behaviors when a child "misbehaves"?

4. To what extent is a gentle parenting approach associated with parent wellbeing?

Because this is the first empirical investigation of the "gentle parenting" approach, we had no guided hypotheses for the third and fourth research questions.

## Method

### Participants

Participants included 100 parents of at least one young child between the ages of 2–7 ($M_{age}$ = 38.58 years, $SD$ = 4.07; 84 females, 16 males). Participants resided in one of three metropolitan areas in the Midwest, Southeast, and Southwest in the US. They primarily identified as White (87%), with 7% identifying as Hispanic, 5% as Asian/Pacific Islander, and 1% as Black or African American. Participants were highly educated: 65% reported a postsecondary degree; 25% had a bachelor's degree; and 10% had an associate's degree, some college, or a high school diploma or equivalent.

### Procedure

The study was approved by the authors' institutional review boards (Macalester College served as the primary IRB of record [#23046]; Rollins College served as oversight IRB). These review boards were guided by the principles of The Belmont Report, federal regulations, state and local laws, and college policies. We distributed a link to a Qualtrics survey (www.qualtrics. com) via email to colleagues and personal acquaintances in various preschools, childcare facilities, and churches in the three metropolitan areas. We included a brief description of the study's goal to explore parenting practices and challenges in 2023 in the wake of the COVID-19 pandemic. Our contacts shared information about the study and the survey link with individuals in their network who met our inclusion criteria of identifying as a parent of a currently 2- to 7-year-old child. Upon opening the survey link, participants provided written informed consent after reading about the study's purpose, the risks and benefits of participation, assurances of the confidentiality of their personal information and responses, the voluntary nature of the study, and their ability to withdraw at any time. The survey took approximately 20 minutes to complete ($M$ = 22.6, $SD$ = 15.4) and participants received a $10 Amazon gift card as compensation within 48 hours. Data collection commenced on July 25th, 2023, and ended on October 26th, 2023, at which time we exported participants' responses to SPSS (quantitative measures) and Excel (qualitative responses) for data cleaning, coding, and analysis. Of the 111 total individuals who completed the survey, we utilized all responses for establishing qualitative coding reliability. To ensure the credibility of responses, we included three validation questions in the survey, which asked participants to select a particular response; 11 respondents failed those questions and thus we deleted their data for our primary analyses.

### Measures

**Parenting approaches.** We measured parenting approaches by presenting participants with a list of 16 parenting adjectives: accepting, affectionate, confrontational, conscious, consistent, expressive, firm, gentle, indulgent, intentional, permissive, reactive, respectful, responsive, strict, and warm. Participants selected any/all adjectives that described "how your primary parental figure(s) approached parenting when you were a young child." In a subsequent question, participants selected from the same list of adjectives to indicate their own personal approach to parenting their 2- to 7-year-old child(ren). For any participants who selected 'gentle' within the list of adjectives to describe their own personal approach to parenting, they were given a follow-up question to describe "what does <u>gentle</u> mean to you as it applies to your personal approach to parenting?"

**Parenting styles.** Parenting styles were measured using 19 items from the Parenting Style Scale [32], which was adapted from Baumrind's original typology. Items were scored on a 5-point Likert scale (1 = Never; 5 = Always) with four items measuring *indulgent* (i.e.,

*permissive*) *parenting* (e.g., "*I make decisions in consultation with my child.*"; ɑ = .72), four items measuring *authoritative parenting* (e.g., "*I give my child reasons for my directions.*"' ɑ = .73), five items measuring *authoritarian parenting* (e.g., "*I have the final say with my child.*"; ɑ = .62) and four items measuring *neglectful parenting* (e.g., "*My child wins arguments with me.*"; ɑ = .64). Items were averaged within each subscale, so individual scores could range from 1 to 5 for each parenting type. Given our interest in comparing gentle parenting to existing parenting styles, we excluded one item on the authoritative subscale that specifically described using a "*gentle manner with my child.*" For the remaining 18 items, we conducted a principal components analysis to confirm the four parenting styles using an oblique rotation (direct oblimin) to simplify the factors and allow factors to be correlated. The resulting structure matrix indicated four clear factors that corresponded with the intended parenting styles, with the exception of one item ("*I tell my child how happy he/she makes me*") that did not load on any of the four factors (factor score < .4). We excluded that item from the indulgent subscale.

**Parental responses to child's "misbehavior".** To assess parental responses to their child (ren)'s misbehavior, parents responded to the prompt: *Think about a recent time when your 2- to 7-year-old child was exhibiting challenging behavior (e.g., 'acting out,' misbehaving, or being disruptive at home or in public). How did you respond in that situation*? We coded participants' narrative responses for categories, and then we analyzed and synthesized those categories on a higher conceptual plane for meaningful groupings.

**Parent wellbeing.** Parent wellbeing was operationalized using 15 items describing *self-efficacy in the parenting role* and *satisfaction with parenting* from the Parenting Sense of Competence (PSOC) scale [33]. Participants indicated their level of agreement with each item by indicating a number between 1 (strongly agree) and 6 (strongly disagree) that best corresponded to their sense of parenting self-efficacy (e.g., "*Being a parent is manageable, and any problems are easily solved*") and satisfaction (e.g., "*I go to bed the same way I wake up in the morning, feeling I have not accomplished a whole lot*"). The 8 efficacy items were reverse scored so that high scores on both subscales indicated positive parental experience. To analyze the two-factor structure of the PSOC, we conducted a principal components analysis with an oblique rotation (direct oblimin) to simplify the factors and allow factors to be correlated. The resulting structure matrix clearly indicated a two factor structure for efficacy and satisfaction. One of the satisfaction items ("My mother/father was better prepared to be a good mother/father than I am,") loaded strongly on the efficacy factor (factor score >.4), so we included the item as part of the efficacy subscale. An additional satisfaction item ("A difficult problem in being a parent is not knowing whether you're doing a good job or a bad job") did not load on either factor so we excluded it from the composite scores. Items were averaged for each subscale to create composite parenting self-efficacy (9 items) and parenting satisfaction (5 items) scores. Cronbach's alphas were adequate (ɑ = .77 and.81 for efficacy and satisfaction, respectively) and similar to those observed with a normative sample in Gilmore and Cuskelly [34].

## Data analyses

To answer our first research question, *How do those who identify as "gentle parents" define such an approach*?, we engaged in a two-step process. First, we filtered the dataset to include only those who selected 'gentle' as an adjective to describe their parenting approach (*n* = 49); within that subsample, we analyzed participants' open-ended descriptions of what 'gentle' meant to them. This analysis was an inductive approach [35] in which we engaged in independent, close readings of the responses and open coding, making notes about any specific themes or descriptors used to illustrate what gentle parenting meant to participants. After discussing the emerging themes identified in the gentle responses, we engaged in an iterative process of (a)

developing and refining coding categories with operational definitions regarding parents' descriptions, (b) continued close readings of participants' gentle parenting descriptions, and (c) ongoing discussion about relevant examples and counterexamples in individual responses.

To further explore the meaning of gentle parenting, we compared whether gentle parents differed from non-gentle parents in terms of the frequency with which they selected specific adjectives (other than gentle) to describe their personal approach to parenting; we employed a series of Pearson Chi-Square tests for such comparisons. We also compared gentle and non-gentle parents on the total number of descriptors they used to describe their parenting approach; we employed an independent samples *t*-test for those comparisons. Finally, we explored the extent to which the number of adjectives used to describe participants' own parenting approach differed in comparison to the number of adjectives they used to describe their parents' approach; we explored these differences among both gentle and non-gentle parents, and we used paired-samples *t*-tests for those comparisons.

To answer our second research question, *How does "gentle parenting" relate to Baumrind's parenting typology*?, we conducted a series of independent samples t-tests to compare differences in authoritative, authoritarian, indulgent, and neglectful parenting between gentle and non-gentle parents. To control for Type I error due to multiple comparisons, we applied a Bonferroni correction for the four pairwise comparisons (authoritative, authoritarian, indulgent, neglectful). Given these comparisons, and our experiment-wise alpha level of 0.05, the corrected alpha was set to.0125 ($p < 0.05/4$).

To answer our third research question, *How do gentle parenting ideologies relate to parenting behaviors when a child "misbehaves"*?, we engaged in an inductive approach [35], closely reading the responses from all participants that described how they responded when their child was "recently exhibiting challenging behavior." This process of inductive analysis involved close readings, open coding, and the creation of emergent themes, which were refined via several rounds of independent coding, discussion of agreement, and refinement of a coding scheme until we achieved strong reliability (Cohen's κ >.80) for each category. We focused primarily on the misbehavior responses of gentle parents, but to explore further our interest in the extent to which gentle parenting is a distinct parenting approach, we compared the misbehavior responses between those identifying as gentle parents and those who did not.

To answer our fourth research question, *To what extent is gentle parenting associated with parent wellbeing*?, we conducted two-tailed, independent-samples t-tests to determine the extent to which mean differences in parenting efficacy and parenting satisfaction differed among those who identified as a gentle parent versus those who did not. To control for Type I error due to multiple comparisons, we applied a Bonferroni correction for the two pairwise comparisons of parenting efficacy and parenting satisfaction. Given these comparisons, and our experiment-wise alpha level of 0.05, the corrected alpha was set to.025 ($p < 0.05/2$). In addition to exploring the quantitative data, we analyzed the qualitative data for emergent themes regarding parental wellbeing, parental stress, or parental support.

## Results

### How do those who identify as "gentle parents" define such an approach?

Through our inductive analysis of the meaning of 'gentle' according to self-identified gentle parents' responses (*n* = 49), we identified three overarching themes that described a gentle parenting approach: *regulating one's own emotions*, *assisting children in regulating their emotions*, and *showing emotional and physical affection* to children. We achieved strong inter-rater reliability for each coding category within this gentle parenting coding scheme (Cohen's κ >.80) [36].

The most prominent theme in gentle parents' responses was about parents' emotion regulation (59.2%). This theme emphasized parents' efforts to stay calm and/or avoid physical punishment or any punitive measures with their child. For example, one gentle parent, a 40-year-old mother of two children, ages 7 months and 3 years, noted that gentle parenting means "being aware of the words I use, the tone of my voice, and taking deep breaths or stating 'I'm frustrated' instead of just showing the emotion." This mother was prioritizing her own emotion regulation in front of her children, clearly preferring to articulate what she feels rather than "just showing the emotion" (e.g., through losing her temper). Gentle parents' descriptions of assisting their children in regulating their emotions was the second-most common theme (40.8%), and it involved validating and listening to children's emotional experiences and/or scaffolding emotions, such that they used their child's emotional experiences as teachable moments for growth and self-understanding. This theme was evident in the response of a 36-year-old mother of two children, ages 1 and 3, who described her gentle parenting approach as "helping my kids understand how their feelings can impact themselves and those around them so that they can build healthy coping skills." Parents' descriptions of showing physical and/or emotional affection to their child was the third-most common theme (26.5%), illustrated through one parent, a 38-year-old mother of three children, ages 3, 6, and 8, who commented that to her, gentle parenting involves "showing them affection that shows I care and love them."

Gentle parenting definitions were often multi-thematic. Thirty-four percent (*n* = 17) of the descriptions of gentle parenting in our sample referenced more than one emergent theme, such as this response from a 37-year-old father of three children, ages 2, 5, and 7 [with **bolded inserts** added to illuminate themes]: "Gentle parenting means mostly maintaining a lower energy level [**parent regulates own emotions**] and being respectful of my children's need for rest and relaxation [**parent regulates child's emotions**]", or this response, from a 39-year-old mother of two children, ages 3 and 5: "Gentle parenting means no harsh words [**parent regulates own emotions**] and soft touches [**shows affection**]." Responses that defined gentle parenting with only one theme (*n* = 32) were still richly descriptive (average word count of single-themed responses was 20 words), such as the following response:

> To me, gentle refers to my response [**parent regulates own emotions**] in frustrating or highly emotional situations with our daughter. We do not hit, spank, demean, or punish our child to make them feel like less of a person. Gentle refers to our discipline techniques and how we respect our child enough to make better decisions on her own. Even if we have to say something many times, we still are not aggressive towards our child to get the point across.
>
> —*42-year-old mother of two children, ages 6 and 10.*

These themes are further illustrated in Table 1.

The multi-thematic nature of gentle parenting also was illustrated in the number of adjectives that gentle parents ascribed to themselves. Among participants who selected gentle to describe their own parenting, 94% also selected affectionate, 88% accepting, 86% warm, 82% intentional, 78% respectful, and 71% conscious. The Pearson Chi-Square tests comparing whether the frequency of these adjectives differed among gentle vs non-gentle parents revealed that gentle parents were statistically more likely to describe themselves as 'expressive' ($rf$ = .57) than were non-gentle parents ($rf$ = .35), $X^2(1, N = 100) = 4.80$, $p = .028$, but they did not differ from non-gentle parents in the frequency with which they selected any other adjectives to describe their approach to parenting.

**Table 1. Themes in the meanings of gentle parenting, according to gentle parents (*n* = 49).**

| Coding Category<br><br>(Inter-rater reliability; Frequency of responses; Percentage of responses) | Operational Definition | Example |
|---|---|---|
| Parent regulates own emotions | | |
| Staying calm (κ = .96; N = 23; 47%) | Parent emphasizes their maintenance of a calm state, such as keeping voice at a low, soft tone; taking deep breaths; taking time to respond; no yelling, screaming, or threats | Having a moderate reaction—never getting too alarmed or being too permissive, always monitoring and adjusting expectations to the needs of the child and environment. |
| Avoiding physical punishment or punitive measures (κ = 1.00; N = 10; 20%) | Parent explicitly states that they refrain from hitting, spanking, or grabbing their child. | To me it means not using time outs or spankings. We choose to be a partner to our child in learning how to be the best human they can be. |
| Parent assists child regulate their emotions | | |
| Validating emotions (κ = 1.00; N = 15; 31%) | Parent attends to, respects, tries to understand, and validates child's feelings/needs. This includes listening with empathy. | Gentle means my children feel that they can come to me with an issue or concern and be received with patience and feel heard. |
| Scaffolding emotions (κ = 1.00; N = 7; 14%) | Parent uses child's emotional experiences as teachable moments for growth and self-understanding. This includes helping child learn coping skills, such as labeling emotions with words, and helping child "get to the root of her feelings." This also includes "processing" with children, and using clear communication with a rationale provided, or explaining the 'why' behind decisions. | Always providing a soft landing pad for him to explore his own thoughts and processes. . .gentle to me really means that he knows that I will consistently guide him without anger or frustration directed towards him as he learns to navigate the world and his place in it. |
| Parent shows affection (κ = 1.00; N = 13; 27%) | Parent shows loving emotional and physical affection to child (e.g., hugs); descriptions of being loving, kind, warm | Soft language, maybe a hug & sitting together for awhile |

The comprehensive nature of gentle parenting was again revealed when we compared gentle and non-gentle parents on the total number of descriptors they used to describe their parenting approach. An independent samples *t*-test indicated that gentle parents selected significantly more adjectives overall to describe their own parenting (*M* = 8.65, *SD* = 2.06) in comparison to non-gentle parents (*M* = 6.94, *SD* = 1.83), *t*(98) = 4.41, *p* < .001. A paired samples *t*-test indicated that gentle parents also provided approximately 50% more adjectives to describe their own parenting approach in comparison to the number of adjectives they used to describe their own parents (*M* = 5.98, *SD* = 2.23), *t*(48) = 9.70, *p* < .001), describing themselves as being more accepting, affectionate, conscious, consistent, expressive, intentional, respectful, and warm, and less indulgent, reactive, and strict (see Fig 1). Importantly, these differences were not unique to gentle parents: non-gentle parents also described their own parenting style with more adjectives in comparison to the number of adjectives they used to describe how they were reared by their parents (*M* = 4.94, *SD* = 2.00), *t*(50) = 7.32, *p* < .001.

## How does gentle parenting relate to Baumrind's parenting typology?

To examine our second research question, we conducted a series of independent samples *t*-tests to compare differences in authoritative, authoritarian, indulgent, and neglectful parenting between gentle and non-gentle parents. There were no statistically significant differences between gentle parents and those who did not identify with this parenting approach in terms of authoritative, indulgent, authoritarian, or neglectful parenting (see Table 2).

## How do gentle parenting ideologies relate to parenting behaviors when a child "misbehaves"?

Similar to the comprehensive definitions of gentle parenting, gentle parents' responses to their child's misbehavior were multifaceted, often illustrating a number of approaches. Through our iterative approach of reading, coding, and creating emergent themes, we established a coding

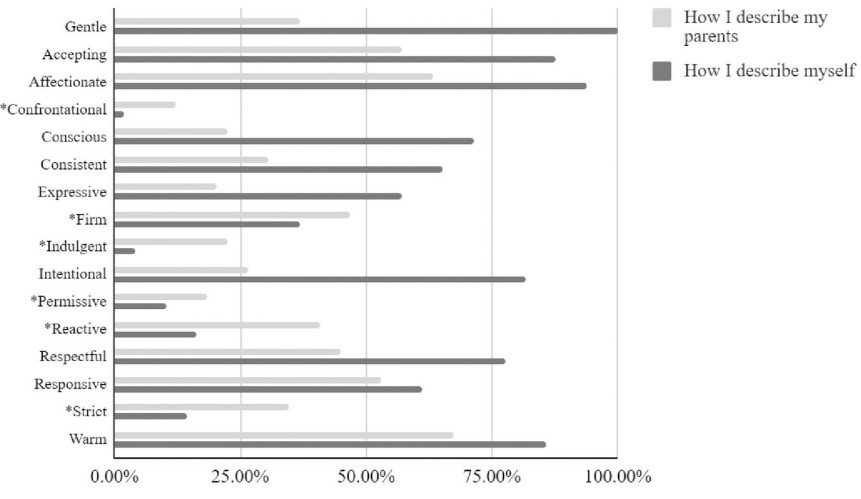

**Fig 1. How do you describe your parenting approach vs. how did your parents raise you? Differences in adjectives among gentle parents (*n* = 49).** *Note.* * indicates a statistically significant (*p* < .05) difference in the percentage of adjectives that gentle parents (*n* = 49) used to describe their own parenting approach versus the number of adjectives used to describe how they were raised by their parents.

scheme consisting of 11 distinct parenting responses to their child's misbehavior, with strong reliability (Cohen's κ >.80) of our interpretations for each category. In addressing their child's misbehavior (e.g., their child throwing a tantrum, being physically aggressive with a sibling or peer, or taking something that was not theirs), gentle parents' responses included showing their child affection, emphasizing a sense of calm, "letting it fly" (i.e., letting their child "feel their feelings"), minimizing hierarchy (i.e., "getting on the child's level"), bargaining, asking their child to stop, punishing their child, "calling out" their child's misbehavior, rationalizing (i.e., explaining why the behavior was inappropriate), redirecting, or disengaging (i.e., the parents would walk away).

On average, gentle parents employed three different responses in these moments of misbehavior from their children (*M* = 2.96, *SD* = 1.33), revealing a multifaceted approach to those moments. The excerpt below illuminates this complex response to misbehavior [**bolded inserts** added to illuminate the tagged code]:

I have noticed that my responses differ in the situation (likely not good). At home—we take quite a bit of time to get low to the ground (at her level), affirm her emotions ("I see you are upset right now") [**minimize hierarchy**], try to understand the cause (if it's not obvious),

**Table 2. Differences between gentle and non-gentle parents in Baumrind's parenting approaches, age, and number of children.**

| Variable | Gentle Parents (*n* = 49) | | Not Gentle (*n* = 51) | | *t*(98) | *p* | Cohen's *d* |
|---|---|---|---|---|---|---|---|
| Parenting Approach | *M* | *SD* | *M* | *SD* | | | |
| Indulgent | 4.20 | 0.47 | 4.03 | 0.53 | 1.65 | .051 | .186 |
| Authoritative | 3.95 | 0.45 | 3.85 | 0.61 | .745 | .458 | .152 |
| Authoritarian | 3.38 | 0.50 | 3.39 | 0.41 | -.504 | .615 | .102 |
| Neglectful | 2.40 | 0.46 | 2.41 | 0.48 | -.131 | .896 | .027 |
| Age | 39.02 | 3.83 | 38.16 | 4.27 | 1.063 | .291 | .213 |
| Number of Children | 2.10 | 0.77 | 2.12 | 0.65 | .109 | .913 | .022 |

try to explain the circumstances (although I read that toddlers can't rationalize so well) [**rationalize**], and sometimes try to let her solve the problem for herself (probably less often), offer acceptable choices for moving forward that result in an acceptable outcome for us ("do you want to do x this way or that way") [**redirect**]. Whenever there is a set-back in this process, we say "do you need a hug? Let's hug it out." [**show affection**] then proceed to the next step.

—42-year-old mother of one child, age 3.

To further explore those responses to children's misbehavior, we categorized the 11 distinct parenting responses into what we perceived to be either parent-directed responses or child-directed responses. Parent-directed responses were those that were guided by the parent and that represented firmer boundaries, or demarcated rules and/or limits, between the parent and child. These responses included punishing, calling them out, rationalizing, redirecting, and disengaging. Child-directed responses were those that were more child-centered and prioritized giving the child more control, power, and/or attention; they also de-emphasized the importance of or minimized the boundaries between the parent and child, reflecting a more democratic parenting approach. These responses included showing affection, emphasizing calm, letting it fly, asking, minimizing hierarchy, and bargaining (see Table 3).

Within each of the misbehavior narratives, participants received a binary score of either 0 or 1 for each of the 11 parenting responses, depending on whether that response was evident in the narrative. We then summed the total number of parent-directed responses and the total number of child-directed responses for each participant. Overall, gentle parents were equally likely to describe parent-directed ($M = 1.37$, $SD = .88$) and child-directed ($M = 1.33$, $SD = 1.07$) responses, $t(48) = .19$, $p = .854$. This finding was distinct from those who did not identify as gentle. Among non-gentle parents, responses were more likely to be parent-directed ($M = 1.49$, $SD = .77$) than child-directed ($M = 1.21$, $SD = .96$), but this difference was not statistically significant, $t(49) = 1.50$, $p = .137$. Among gentle parents, there was much variation in the extent to which they favored a parent-directed or child-directed approach. To capture this variation, we computed a difference score, subtracting the number of child-directed responses from the number of parent-directed responses, such that positive scores indicated participants' inclusion of more parent-directed responses in their misbehavior narratives, negative scores indicated more child-directed responses, and scores of 0 indicated that self-identified gentle parents described an equal mix of parent- and child-directed responses. Consistent with findings that gentle parenting was multi-thematic and multifaceted, there was variation in gentle parents' difference scores indicating that gentle parenting is practiced on a spectrum. Difference scores ranged from -3 (higher on child-directed misbehavior responses) to 3 (higher on parent-directed responses). Most gentle parents (67%) described a combination of parent-directed and child-directed responses. Twenty gentle parents (40.82%) had negative difference scores indicating misbehavior response descriptions that included more child-directed responses, 18 gentle parents (36.73%) had positive difference scores, indicating descriptions with more parent-directed responses, and 11 gentle parents (22.45%) had scores of 0 indicating an equal number of parent- and child-directed responses described in their misbehavior narratives. In contrast, 16 non-gentle parents (31.37%) described more child-directed responses, 26 (51.01%) described more parent-directed responses, and only nine (17.64%) described an equal number of parent- and child-directed responses.

The following narrative illustrates a misbehavior response from a gentle parent who described using only child-directed strategies (difference score = -3) [**bolded inserts** added to illuminate the tagged child-directed code]:

**Table 3. Themes in gentle parents' (*n* = 49) narratives of response to child's misbehavior.**

| Coding Category (Inter-rater reliability; frequency; percentage) | Definition | Example |
|---|---|---|
| PARENT-DIRECTED RESPONSES | | |
| Punish (κ = .96; N = 15; 31%) | Either taking away a privilege (e.g., screen time or treats) or administering unpleasant stimuli (e.g., yelling, grabbing their arm, forcing an apology) in response to misbehavior. Allowing a natural consequence that is unpleasant for the child. | I tell them to knock it off.<br><br>[I] separate and punish them, |
| Call them out (κ = 1.00; N = 15; 31%) | Telling child that their behavior is not appropriate. Directly addresses the misbehavior. | I tell them that that behavior is unacceptable.<br><br>I said firmly that her behavior wasn't okay. (Something like "This behavior needs to stop. It's hurtful and not helpful.") |
| Rationalize (κ = .96; N = 24; 49%) | Explaining the reasons why their behavior was not okay. This reasoning can occur during misbehavior and sometimes afterwards when processing/debriefing the incident. | Try to explain the circumstances ("although I read that toddlers can't rationalize so well").<br><br>I explain why changing his behavior was important. I tried to explain it in a way that reflected the importance in relation to him and those around him. |
| Redirect (κ = 1.00; N = 10; 20%) | Attempting to re-channel children's 'acting out' or misbehavior to something more productive/healthy; offering a positive alternative to misbehavior that does not involve bribery. | I tried to redirect to do something positive with her like have some cuddle time and read a book together.<br><br>I offer and model a replacement behavior. |
| Disengage (κ = .85; N = 3; 6%) | Explicitly stepping back from or leaving an emotionally intense scenario in an effort to stay calm. Usually manifested in walking away and/or taking a deep breath. Stating the need to "hold/honor a boundary" around their own wellbeing and prioritizing that boundary over their child's demands. | I had to walk away and disengage to send the message that the extreme behavior will not get a response.<br><br>I usually try to take a second to compose myself |
| CHILD-CENTERED OR CO-CONSTRUCTED RESPONSES | | |
| Show affection (κ = 1.00; N = 10; 20%) | Providing any physical affection (e.g., hugging) or emotional affection (e.g., "I tell her I love her"). | Whenever there is a set-back in this process, we say "do you need a hug?"<br><br>Reminded her I loved her. |
| Emphasize calm (κ = .96; N = 14; 26%) | Trying to encourage/assist their child in calming their bodies when they are upset. Working towards a calm goal-state in the child. | I usually try to help my child regain her calm (to regulate, suggest some deep breathing or naming the big emotion) and have her use more appropriate behaviors (calm tone or "tell me in words please so I may understand you" type of directives) to come up with a choice/resolution. |
| Let it fly (κ = 1.00; N = 19; 39%) | Letting the child "feel their feelings" without interruption or interference. | Waited until his fit was over.<br><br>I let them have their feelings about it. |
| Minimize hierarchy (κ = 1.00; N = 12; 25%) | Attempting to "get on child's level", either physically (e.g., bringing body at the same height of child, sitting with them) or emotionally (e.g., acknowledging feelings, validating feelings). | Sat my child down on a nearby chair, knelt down so we were on the same level, made physical contact & eye contact<br><br>I validated their experience |
| Bargain (κ = 1.00; N = 5; 10%) | Negotiating with or bribing the child (e.g., giving food or screens) under the assumption that their "misbehavior" stemmed from an unmet need. This can also be preemptive bribes to help elicit desired behavior. | Asked them what they needed in the moment, tried to provide that if it was appropriate to the situation. |
| Ask (κ = 1.00; N = 5; 10%) | Asking child to stop misbehaving (e.g., can you use your words?). | Typically we ask several times to stop.<br><br>Asked nicely for them to please stop said behavior (being mean to little brother, not listening when I ask him to do something, etc.) |

We try our best to stay calm when children are "acting out." We let them feel their feelings [**let it fly**], and if they're really upset, go with them to another room to calm their bodies down [**emphasize calm**]. Later in the day when they are calmed down, we talk through what happened and why they felt that way [**minimize hierarchy**], what they could do

instead if they got physical. Sometimes we don't react the way we want as parents! We never yell, but are sometimes more firm or say things we didn't mean to say. In those situations, we apologize and talk.

*—35-year-old mother of two children, ages 1 and 4.*

Throughout this mother's response, the emphasis is not on parent-governed rules or expectations, but rather, the focus is on the child's feelings, needs, and abilities to (later) process whatever misbehavior had occurred. Parent-directed boundaries are not emphasized in these child-directed responses.

Contrasting those child-directed strategies were those from parents who endorsed a parent-directed response to misbehavior, such as the following (difference score = + 3) [**bolded inserts** added to illuminate the tagged parent-directed code]:

Last night, she was in the bathtub and kicking the water so that it was splashing out of the tub. I explained to her that she could kick smaller in the tub, but big kicks are for the pool [**call them out**]. Big kicks in the tub is getting water everywhere. I explained that if she kept splashing, bathtime would be over [**rationalize**]. She was laughing and continued to kick water out of the tub, so I started draining the water, picked her out of the tub, and gave her the towel [**punish**]. She got back in the tub, and I repeated, no, bathtime is over, picked her out of the tub, and put the towel around her. She got upset and was crying. I think I acknowledged that it's hard, but that she needs to do little kicks in the tub.

*—37-year-old mother of two children, ages 1 and 3.*

Throughout this mother's response, the emphasis is on boundaries—on clearly stated rules and/or limits, set and maintained by the parent—even when there was clear disagreement about those rules from the child.

Situated between those two distinct approaches to misbehavior were the gentle parents who equally referenced a combination of parent- and child-directed responses when they discussed how they responded to their child(ren)'s misbehavior. The following response generated a difference score of zero, since it included both punishing (a parent-directed response) and bargaining (a child-directed response) [**bolded inserts** added to illuminate the tagged parent- and child-directed code]:

[I] Tell the child how to act, walking feet, inside voices. Since my child is young if we go somewhere I feel a behavior might happen I remind them of expected behavior at that location and if they do so what they will earn [**bargain**]. When they don't have expected behavior they don't get to earn privileges [**punish**].

*—33-year-old mother of one child, age 3.*

## To what extent is a gentle parenting approach associated with parent wellbeing?

In the current study, we had no directional hypotheses about whether gentle parents would differ from non-gentle parents in terms of their parenting self-efficacy or satisfaction. To explore this potential variance, we conducted two-tailed, independent-samples *t*-tests to determine the extent to which mean differences in parenting efficacy and parenting satisfaction

differed among those who identified as a gentle parent versus those who did not. Parenting self-efficacy did not differ significantly among gentle- versus non-gentle parents ($t(98)$ = 1.48, $p$ = .142), although mean scores were higher among gentle- ($M$ = 4.32, $SD$ = 0.61) compared to non-gentle parents ($M$ = 4.13, $SD$ = 0.65). Gentle parents reported more satisfaction ($M$ = 4.15, $SD$ = 0.96) than did non-gentle parents ($M$ = 3.73, $SD$ = 0.92), although the difference did not reach statistical significance with the Bonferroni correction, $t(98)$ = 2.26, $p$ = 0.026.

In slight contrast to these quantitative findings was a theme that arose from the qualitative data. Within the narratives gentle parents provided, they often reflected on their wellbeing, openly questioning whether they were "doing it [parenting] right." To gain a better understanding of these self-evaluations, we studied the parenting approach descriptions parents provided and we identified additional, similar, evaluative reflections. In response to the question about parenting approaches, one gentle parent, a 38-year-old mother of three children, wrote simply: "I'm hanging on for dear life." Another, a 43-year-old mother of two children, ages 3 and 5, wrote that "I confess I feel I have no idea what I'm doing much of the time." Following this emergent theme, we read across all gentle parents' narrative responses to identify any utterances of what we called *self-critique*, which we defined as moments of parenting uncertainty, insecurity, and/or doubts about personal parenting efforts/attempts. Overall, 34.7% of gentle parents' responses about their parenting approach or about how they handled their children's misbehavior contained unprompted elicitations of self-critique. Below are several examples of gentle parents' self-critique about their parenting approach or response to their child's misbehavior:

> I would describe it [my parenting approach] as still exploring. I have read ALL the parenting books, listened to the podcasts, followed the socials. . . and can't say I feel I've found any one "philosophy" that we adhere to all the time. . .I don't really feel like I know what I'm doing—so still exploring what "my approach" is.
>
> —*38-year-old mother of two children, ages 5 and 7.*
>
> As a stay at home mother I get easily overstimulated and overwhelmed all day every day. I very much look forward to nap time and bedtime because those are pretty much the only times I get to be alone.
>
> —*36-year-old mother of two children, ages 1 and 4.*
>
> In public. . .we tend to rush through the normal process, skip steps, with varying degrees of success. When/if they don't work, we are more likely to transition into bribing/negotiating —which we are aware are not recommended approaches. We have observed that we use food too often as the carrot, which we need to work on.
>
> —*42-year-old mother of one child, age 3.*

Because of the prevalence of the self-critique theme in gentle parents' responses, we explored the extent to which parenting efficacy and parenting satisfaction differed among those gentle parents who self-critiqued versus those who did not. Among all gentle parents in our sample, parenting self-efficacy among those who engaged in self-critique ($M$ = 3.96, $SD$ = 0.46) was significantly lower ($t(47)$ = 2.69, $p$ = .01) than the parenting self-efficacy of those who did not engage in self-critique ($M$ = 4.44, $SD$ = 0.65). Parenting satisfaction among those who engaged in self-critique ($M$ = 3.94, $SD$ = 0.90) also was lower than the parenting satisfaction of those who did not engage in self-critique ($M$ = 4.13, $SD$ = 0.70), but this difference was not statistically significant, $t(47)$ = 0.79, $p$ = .434.

## Discussion

Parenting young children has always been hard, but evidence suggests that it might be getting harder [10]. The pressures to fulfill exacting parenting standards, coupled with the information overload on social media about the right or wrong ways to care for children, has left many parents questioning their moment-to-moment interactions with their family and leaving them with feelings of burnout [2]. Such feelings may have emerged as many caregivers increased their time spent in active parenting (i.e., more play/learning activities) and decreased their time spent disciplining their young children during the lockdown period, a trend that has, notably, emerged across the income spectrum [1]. Within this time of parenting angst, the idea of "gentle parenting" has gained popularity, especially on social media, as a promising new approach to raise happier, healthier children. Since the idea of gentle parenting was first introduced in 2015 by British author and parent Sarah Ockwell-Smith, coverage of this approach has skyrocketed online. At the time of this publication, a Google search on "gentle parenting" generates 1.68 million hits. That number will undoubtedly continue to climb.

Despite the popularity of this approach, there has been no empirical research about what gentle parenting actually means among those who endorse this approach; no systematic investigations to interrogate the extent to which gentle parenting is, in fact, a distinct approach from other, established typologies of parenting; no explorations to determine how gentle parents respond to challenging moments with their children; and equally important, no pointed inquiries to consider whether gentle parents are faring any better than the rest of us. This study represents a preliminary step to address those important questions.

## The meaning of 'gentle parenting' according to gentle parents

Forty-nine percent of our sample described their personal approach to parenting as 'gentle'. These gentle parents ranged from 32 to 51 years of age, and they were no more or less likely to have additional children than non-gentle parents. At least among the three US urban populations from which we sampled, gentle parenting appears to be a widely endorsed approach to parenting young children among Millennials and Gen-Xers. Our analyses suggest that gentle parenting is nuanced and multi-thematic, especially in comparison to the ways these gentle parents were raised when they were children, but it is primarily an approach that prioritizes parents' and children's emotion regulation and a high amount of parental affection. Almost 60% of self-identified gentle parents defined 'gentle' as regulating their own emotions, which involved maintaining a sense of calmness and a cool head, as well as avoiding any punitive actions, such as yelling or spanking, when it came to parenting their young children. Consistent with our expectations, this theme is similar to contemporary positive parenting movements that emphasize the importance of caregivers remaining 'mindful' [25] and 'conscious' [26]. Parent emotion expression and regulation have long been considered integral to young children's developing social and emotional competence [37]. Parents provide the primary context in which children first learn about healthy (and unhealthy) emotion expression and management [38], and evidence suggests that parents with better emotion regulation skills exhibit more positive parenting behaviors and have children with better emotion regulation and fewer internalizing symptoms [39]. Close to 41% of the self-identified gentle parents in our sample also defined 'gentle' as assisting their children with emotion regulation. These parents emphasized the importance of validating their child's emotions through listening with empathy, understanding, and respect, as well as using their child's emotional experiences as opportunities for nurturing intimacy and teaching coping skills and problem-solving. Both of these themes are consistent with Gottman and colleagues' [22] model of emotion coaching, which affirms the role of caregivers' empathetic engagement and awareness of emotions in

themselves, as well their critical role in acknowledging and scaffolding their children's emotional expressions. A third theme mentioned by around a quarter of gentle parents focused on showing physical and/or emotional affection to their child. These descriptions of hugging their child and being loving, kind, and warm are consistent with what positive parenting endorses [e.g., 40], and such maternal and paternal behaviors are largely positive in terms of their correlates with child outcomes [41, 42].

## Associations between gentle parenting and Baumrind's typology

As to whether or not gentle parenting is truly a unique approach or is, figuratively, "old wine in new skins," the evidence from this study suggests that, consistent with our expectations, gentle parenting is similar to Baumrind's (1966) authoritative and indulgent parenting styles; based on the adjectives gentle parents used to describe their approach, it also seems to be similar to the positive parenting model [40, 41]. Still, this approach likely merits its own qualitatively distinct category in that it emphasizes *boundaries* as a novel alternative to discipline. In our analyses of gentle parents' responses to their children's misbehavior, we noted that 28% included a reference to boundaries. "We try to hold consistent boundaries," wrote a 39-year-old mother of two kids, ages 3 and 11. "We focus a lot on communicating boundaries," wrote a 35-year-old father of one child, age 5. This emphasis on boundaries is distinct from other parenting approaches, but interestingly, gentle parents' behaviors—or at least, their self-reports of their behaviors—showed a great deal of diversity in terms of how those boundaries were carried out in moments of their children's misbehavior. About 37% of gentle parents in our sample demonstrated fairly clear, demarcated rules or boundaries with their children (what we referred to as a parent-directed approach); about 40% illustrated a more child-directed, democratic approach, with less-clear boundaries; and the remainder showed a combination of approaches, some that minimized boundaries and others that bolstered them. This diversity underscores the notion that gentle parenting is not a uniformly applied approach, but rather a multi-thematic philosophy and a multifaceted practice that is best conceptualized on a spectrum.

**"I do the opposite of my parents".** A common sentiment expressed among the gentle parents in our sample was that they would do a better job at rearing their young children than their own parents had done with them. In comparison to their descriptions of their own parents, gentle parents described themselves as more accepting, affectionate, conscious, expressive, respectful, responsive, warm, and intentional. Indeed, "more, more, more" was the theme of these gentle parents. On average, they reported 50% more adjectives in describing their own parenting style than they used to describe how they were raised by their parents. These gentle parents would do more for their children. They would *be* more.

For some in our sample, the emphasis seemed to be about undoing what they perceived to be harmful parenting practices that they had experienced when they were children. For example, one gentle parent, a 43-year-old mother of three children, ages 8, 4, and 8 months, wrote that her approach to parenting was, quite simply, to "do the opposite of my parents. No spanking or physical punishment." This desire to not only "do more" for their children but to "do differently" can be seen in the adjectives gentle parents selected to describe themselves versus their parents. Eighty-two percent of gentle parents in our sample described themselves as "intentional," a stark contrast from the "reactive" descriptor that 42% of them ascribed to their parents. These findings are aligned with the recent report from the Pew Research Center [10], which revealed that 43% of the sampled parents wanted to raise their children differently than they had been raised, specifically with more love and compassion. This [perception of an] inter-generational divide in parenting values is meaningful, and represents an under-explored

area of research in developmental psychology. Future research should examine the relationships between self-identified gentle parents and their own parents to further elucidate the findings in our study that a subset of gentle parents were rejecting the reactive manner in which they were parented.

**Gentle parents are not-so-gentle on themselves.**   As to how these gentle parents are doing, our data, again, portray a nuanced portrait. Overall, the gentle parents in this study reported high levels of parenting self-efficacy and satisfaction, suggesting that on the whole, they are doing quite well. But the emergent theme of self-critique, expressed by over one-third of gentle parents, and the findings that, among those self-critical gentle parents, the levels of self-efficacy were significantly lower, illuminates the need for more explorations and more support of these parents. One of the gentle parents in our sample, a 40-year-old mother of two children, wrote that her approach to parenting is about "Trying to remain calm. . .but I do reach my limit sometimes." Gentle parenting seems to represent an approach that is extraordinarily gentle for the children, but perhaps not-so-gentle for the parents themselves. Future research would benefit from examining gentle parenting in association with additional measures of parent wellbeing, including parenting stress and perfectionism.

**Limitations and future research directions.**   The limitations of this study include the relatively small, homogeneous sample. We wonder: Is gentle parenting a predominantly White, high socioeconomic status approach to parenting among mothers? Is gentle parenting a stressful experience (at least for some) because of its ideologies, or because of other, unexplored variables? Future research is merited with a more diverse sample, one that accounts for both parents' and child(ren)'s age and gender, the number of children in the family, and family structure, to explore the extent to which gentle parenting is defined and/or practiced similarly among other demographic groups. The current study also was limited in its means of data collection. Although our survey data were rich in detail, it was limited in that it only gathered parents' self-reports on their "general" responses, rather than "child-specific" responses; this approach may not reflect parents' authentic behaviors with their child(ren), and it may have masked important variability among parents who behave in specific ways with one child versus another. Observational data of parent-child interactions are needed, ideally in naturalistic settings. Finally, the current study was limited in its single-time-point assessment of gentle parenting. Especially considering that many gentle parents in this study were self-critical—questioning the merits of this approach and even "overwhelmed all day every day"—longitudinal research is needed on the sustainability of this approach for the parents, as well as its potential impact on child development.

**Conclusions and practical implications.**   Guidance about the "best" ways to rear small children has been around for a long time. What seems to be unique about the gentle parenting movement is that it has not been presented or advocated by scholars of human development; rather, it has largely been the product of social media. Considering that parents are increasingly stressed or burned out by their caregiving responsibilities [12], it is imperative that evidence-based guidance is made available to those who are interested in gentle parenting. What does this approach entail? How is it related to other parenting approaches? Is it a sustainable approach for caregivers? These are empirical questions, and they deserve empirical answers. The current study represents the first systematic investigation of a parenting approach widely endorsed by contemporary parents of young children.

## Supporting information

**S1 Data.**
(SAV)

## Author Contributions

**Conceptualization:** Anne E. Pezalla, Alice J. Davidson.

**Formal analysis:** Anne E. Pezalla, Alice J. Davidson.

**Funding acquisition:** Anne E. Pezalla.

**Investigation:** Anne E. Pezalla, Alice J. Davidson.

**Methodology:** Anne E. Pezalla, Alice J. Davidson.

**Resources:** Anne E. Pezalla, Alice J. Davidson.

**Software:** Anne E. Pezalla.

**Writing – original draft:** Anne E. Pezalla, Alice J. Davidson.

**Writing – review & editing:** Anne E. Pezalla, Alice J. Davidson.

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
