## [Decision Letter · Decision Letter 0]

13 May 2024

PONE-D-24-07088“Trying to Remain Calm…But I do Reach my Limit Sometimes”: An Exploration of the Meaning of Gentle ParentingPLOS ONE

Dear Dr. Pezalla,

Thank you for submitting your manuscript to PLOS ONE. After careful consideration, we feel that it has merit but does not fully meet PLOS ONE’s publication criteria as it currently stands. Therefore, we invite you to submit a revised version of the manuscript that addresses the points raised during the review process.

Thanks for submitting the paper and allowing us to review it. Reviewer number 1 has valid and important comments that will make your paper much stronger. It is an interesting paper and I think it should be published with some clearing up of things like Bonferroni corrections, adding a data analytic section. In my late post doc at Johns Hokpins they taught me that--not in my psych degree but it honestly doesn't help your reader. 

We look forward to receiving your revised manuscript.

Kind regards,

Mary Diane Clark, PhD

Academic Editor

PLOS ONE

Journal Requirements:

2. Thank you for stating the following financial disclosure: "Annie Pezalla received a $1250 faculty fund from Macalester College for this study. She used those funds to compensate participants."  

Additional Editor Comments:

Lucky you---the deadly split--accept it is great and you need to clean this up. I have looked a the comments, they are good comments and I think fairly easy to correct. Can you please do that so we can see if that reviewer would be satisfied?

Isn't this always fun?

Reviewers' comments:

Reviewer's Responses to Questions

**Comments to the Author**

1. Is the manuscript technically sound, and do the data support the conclusions?

Reviewer #1: Yes

Reviewer #2: Yes

2. Has the statistical analysis been performed appropriately and rigorously? 

Reviewer #1: No

Reviewer #2: Yes

3. Have the authors made all data underlying the findings in their manuscript fully available?

Reviewer #1: Yes

Reviewer #2: Yes

4. Is the manuscript presented in an intelligible fashion and written in standard English?

Reviewer #1: Yes

Reviewer #2: Yes

5. Review Comments to the Author

Reviewer #1: Reviewer report

Overall, this study tackles an important topic. Indeed, in the wake of social media parenting, parents have been at times receiving mixed information about what is considered the best approach to parenting and managing difficult behavior in children. I believe this work is important but requires review before being published in this esteem journal. As detailed below, the authors should be more specific in explaining their analysis and adjust for errors, with some sections needing restructuring. The discussion merits to be reviewed to provide a more in-depth explanation of the interesting findings.

More detailed comments

Introduction

1. The introduction is well-written but it would be helpful to have hypotheses or expectations rather than research questions solely, especially that the analyses involves both quantitative and qualitative analyses.

Method

2. Were there any exclusion criteria? You mention in the procedure that “Of the 111 total individuals who completed the survey, we utilized all responses for establishing qualitative coding reliability, but then deleted responses for 11 respondents who failed three validation questions before we conducted our primary analyses.”, what were those? These should be included as inclusion/exclusion criteria

3. Also it is standard practice to describe the measures before summarizing the procedure. I suggest these sections are switched in order.

4. You also mention that “Participants selected from the same list of adjectives to indicate their personal approach to parenting their 2- to 7-year-old child(ren).” Does that mean that parents could report on more than one child? What if their approach to parenting is not “general” but rather “child-specific,” in other their answers might differ when asked to think about different children.

5. I would create a separate “data analysis” section detailing the process of qualitative analysis rather than having it in the section describing the measures. This will make it clear to the reader how you will be using both types of analyses. As it stands it is not clear what the hypotheses are, thus unclear which inferential statistics you will be using.

6. Could you please further expand on how you adapted the Parenting Style Scale to only use 17 items? You cite Saunders et al. (2012) as your reference, I am assuming because they also adapted 22 items of Baumrind original scale; however they described the analysis run to ensure validity and reliability of the factor analysis. I do not see this in your manuscript.

7. Same comment for the PSOC which usually comprises 17 items.

8. Overall this section merits restructuring to have separate sections for: participants, measures (qualitative and quantitative, procedure, data analysis/coding

Results

9. When running so many t-tests you should correct for type I error.

10. “Gentle parents were statistically more likely to describe themselves as ‘expressive’ (rf = .57) than non-gentle parents were (rf = .35), X2(1, N = 100) = 4.80, p = .028, but they did not differ from non-gentle parents in the frequency with which they selected any other adjectives to describe their approach to parenting.” – which tests were used?

11. Same for the next set of results

12. How Does Gentle Parenting Relate to Other Established Parenting Typologies? – you ran t-tests to compare Baumrind’s parenting styles scores across the 2 groups gentle vs non gentle. However, was there a factor structure conducted for that scale after removing some of the items? If the 4-factor structure does not stand, then we cannot use these 4 parenting styles as a basis.

13. “To further explore those responses to children’s misbehavior, we categorized the 12 distinct parenting responses into what we perceived to be either parent-directed responses or child-directed responses” – how did you come up with 12? What was the coding system?

14. “In response to the question about parenting approaches, one gentle parent, a 38-year-old mother of three children, wrote simply: “I’m hanging on for dear life.” Another, a 43-year-old mother of two children, ages 3 and 5, wrote that “I confess I feel I have no idea what I'm doing much of the time.” – I cannot but notice the age of both mothers, I think it is important to account for some many confounding variables, including age, in order to answer the question “How are Gentle Parents Doing?”

Discussion

15. You start your discussion bringing in the COVID-19 pandemic and lockdown; however, you collected data in 2023. Some parents did not yet have their kids in 2019-2020 (lockdown years) so I am not sure how relevant this reference is. Many other factors have affected “parents questioning their moment-to-moment interactions with their family and leaving them with feelings of burnout,” such as the worldwide economic crisis, social media pressure about parenting pressures, more so than COVID-19

16. Given that the authors differentiated between positive and gentle parenting in the introduction, it would be interesting to contrast the two in the discussion. In other word, what is the added value of the themes you highlighted in your results?

17. Comparing gentle parenting results to Baumrind’s typology merits further expansion. The t-test analysis you conducted does not allow for the conclusion you made. It might actually be interesting to run a factor analysis on that questionnaire and then try to map the gentle parenting onto the factor structure you get.

18. Overall, the discussion should be reviewed to include possible confounding variables such as age, number of children in the family, family structure, couple level of support, employment history among others. The results are indeed very interesting but the discussion seems quite superficial.

Reviewer #2: The manuscript is incredibly well-written. It is clear that the authors were mindful of the research and its implications using both the quantitative data and the qualitative data in a back-and-forth display to reach conclusions and consensus. This was done extremely well.

6. PLOS authors have the option to publish the peer review history of their article (what does this mean?). If published, this will include your full peer review and any attached files.

Reviewer #1: No

Reviewer #2: **Yes: **Kyle Jackson

---

## [Author Response · Author response to Decision Letter 0]

24 May 2024

Within this submission, we have supplied the following:

A rebuttal letter that responds to each point raised by our reviewers. Per the instructions we received with this feedback, we uplaoded this letter as a separate file labeled 'Response to Reviewers'.

A marked-up copy of our manuscript that highlights the changes we made to the original version. We uploaded this as a separate file labeled 'Pezalla - Davidson - PLOS - Track Changes'.

An unmarked version of our revised paper without tracked changes. We uploaded this as a separate file labeled 'Pezalla - Davidson - PLOS - Clean'.

Thank you!

~Annie

---

## [Decision Letter · Decision Letter 1]

4 Jun 2024

PONE-D-24-07088R1“Trying to Remain Calm…But I do Reach my Limit Sometimes”: An Exploration of the Meaning of Gentle ParentingPLOS ONE

Dear Dr. Pezalla,

Thank you for submitting your manuscript to PLOS ONE. After careful consideration, we feel that it has merit but does not fully meet PLOS ONE’s publication criteria as it currently stands. Therefore, we invite you to submit a revised version of the manuscript that addresses the points raised during the review process. I have tried to be clear in the attachment of how method and results seem to run together. You need an overall data analytic section (as was suggested by Reviewer 1 in the first review) Please include all analysis and make this a map so that the reader knows what is coming next.   If you have questions please ask me. 

We look forward to receiving your revised manuscript.

Kind regards,

Mary Diane Clark, PhD

Academic Editor

PLOS ONE

Journal Requirements:

Additional Editor Comments

Thank you for t his revision. However, I find it more difficult to follow. You need a map of what is coming as I was confuses. You need to have a data analytic section that explains ALL of the analysis you will do. All of a sudden I have Chi ares and t tests.

Tables 1 and 2 are results -- not sure why they are in the method. I see you did reliability measures but I am not sure why they are there.

I have made comments on the second version where the changes show---if you have questions please email me.

Reviewers' comments:

Reviewer's Responses to Questions

**Comments to the Author**

1. If the authors have adequately addressed your comments raised in a previous round of review and you feel that this manuscript is now acceptable for publication, you may indicate that here to bypass the “Comments to the Author” section, enter your conflict of interest statement in the “Confidential to Editor” section, and submit your "Accept" recommendation.

Reviewer #1: All comments have been addressed

2. Is the manuscript technically sound, and do the data support the conclusions?

Reviewer #1: Yes

3. Has the statistical analysis been performed appropriately and rigorously? 

Reviewer #1: Yes

4. Have the authors made all data underlying the findings in their manuscript fully available?

Reviewer #1: Yes

5. Is the manuscript presented in an intelligible fashion and written in standard English?

Reviewer #1: Yes

6. Review Comments to the Author

Reviewer #1: Thank you for your thorough review of the manuscript and addressing all comments.

I have no further concerns

7. PLOS authors have the option to publish the peer review history of their article (what does this mean?). If published, this will include your full peer review and any attached files.

Reviewer #1: No

---

## [Author Response · Author response to Decision Letter 1]

13 Jun 2024

Thank you for the feedback on our last revised manuscript. Per your requirements, we have revised our Data Analysis section for more clarity on how we analyzed the data for each research question. Now, when readers "arrive" at our Results section, there are no surprises as to how we produced the findings.

---

## [Editor Report · Decision Letter 2]

24 Jun 2024

PONE-D-24-07088R2“Trying to Remain Calm…But I do Reach my Limit Sometimes”: An Exploration of the Meaning of Gentle ParentingPLOS ONE

Dear Dr. Pezalla,

Thank you for submitting your manuscript to PLOS ONE. After careful consideration, we feel that it has merit but does not fully meet PLOS ONE’s publication criteria as it currently stands. Therefore, we invite you to submit a revised version of the manuscript that addresses the points raised during the review process.

There have been many improvements to the paper and it only needs a few more clarifications to be ready to approve. I have made comments in the section below on how to clarify the remaining few items. Do complete the remaining analysis between gentle and non-gentle in terms of how they discipline their children.  I was a bit surprised at some of the comments and it would be important to be sure that they are or are not different. 

We look forward to receiving your revised manuscript.

Kind regards,

Mary Diane Clark, PhD

Academic Editor

PLOS ONE

Journal Requirements:

Additional Editor Comments:

This version is so much easier to read. Given that Plos One doesn't have copy editors, I have gone thru and would like to see some minor changes.

The most problematic for me is the lack of a comparison to "non-gentle" parents in the use of how they parent. Many of the comments were not gentle and I wonder if this is a social media driven issue.

then the last comment here -- you are comparing two things but I am not sure what they are.

Pages 19 and 20 have the same quote to show emotional regulation and only one theme—can you find a different quote?

Then you use it again in the table

Seems like you should be able to pull out some variety.

Also, after a direct quote, it helps to have a concluding sentence to end the paragraph.

Page 22--- selected affectionate, 88% selected accepting, 86% selected warm, 82% intentional, 78% respectful, and 71% selected conscious.

Only need selected the first time

Table 2. Differences Between Gentle and Non-Gentle Parents in Baumrind’s Parenting Approaches, Age, and Number of Children.

Wouldn’t it be better to use a multiple regression, rather than independent t tests? Age and number of children do not add to the variance between the two groups ----however, there are so few differences you would end up reporting no difference which is what you get anyway

However please take out

Compared to those who did not identify as gentle in their parenting approach, gentle parents had higher scores on authoritative and indulgent parenting, and lower scores on authoritarian and neglectful parenting, although differences did not reach statistical significance with the Bonferroni corrections (p < .0125) (see Table 2).

This part simple reflects variation and I do not think it really reflects true differences—you could add it to future research is you think it might be important

How do gentle parenting ideologies relate to parenting behaviors when a child "misbehaves"?

You don’t name the third behavior

Then it seems that many of these types of behaviors are not “gentle parenting”------they lose it, the yell. Are you sure that you have defined a type?

Did these types different between the two groups? I think you need to check this[---what is the simply add the word gentle as it is popular?

Among all gentle parents in our sample, parenting self-efficacy among those who engaged in self-critique (M = 3.96, SD = 0.46) was significantly lower (t(47) = 2.69, p = .01) than the parenting self-efficacy of those who did not engage in self-critique (M = 4.44, SD = 0.65). Parenting satisfaction among those who engaged in self-critique (M = 3.94, SD = 0.90) also was lower than the parenting satisfaction of those who did not engage in self-critique (M = 4.13, SD = 0.70), but this difference was not statistically significant, t(47) = 0.79, p = .434)

How are these different?

Overall, this version is much better and more organized--thank you for the effort.

---

## [Author Response · Author response to Decision Letter 2]

29 Jun 2024

All requested changes have been clearly explained in our "Response to Reviewers 6.29" file, which we have attached within this submission. Thank you!

---

## [Editor Report · Decision Letter 3]

5 Jul 2024

“Trying to Remain Calm…But I do Reach my Limit Sometimes”: An Exploration of the Meaning of Gentle Parenting

PONE-D-24-07088R3

Dear Dr. Pezalla,

We’re pleased to inform you that your manuscript has been judged scientifically suitable for publication and will be formally accepted for publication once it meets all outstanding technical requirements.

Kind regards,

Mary Diane Clark, PhD

Academic Editor

PLOS ONE

Additional Editor Comments (optional):

Thank you for the changes you have made in this 3rd manuscript. Congrats on the clarifications
---

## [Editor Report · Acceptance letter]

9 Jul 2024

PONE-D-24-07088R3 

PLOS ONE

Dear Dr. Pezalla, 

I'm pleased to inform you that your manuscript has been deemed suitable for publication in PLOS ONE. Congratulations! Your manuscript is now being handed over to our production team.

Kind regards, 

on behalf of

Dr. Mary Diane Clark 

Academic Editor

PLOS ONE